# Fecal Microbiota Transplantation in Decompensated Cirrhosis: A Systematic Review on Safety and Efficacy

**DOI:** 10.3390/antibiotics11070838

**Published:** 2022-06-23

**Authors:** Annie S. Hong, Kyaw Min Tun, Jenny M. Hong, Kavita Batra, Gordon Ohning

**Affiliations:** 1Department of Gastroenterology, Kirk Kerkorian School of Medicine, University of Nevada, Las Vegas, NV 89102, USA; gordon.ohning@unlv.edu; 2Department of Internal Medicine, Kirk Kerkorian School of Medicine, University of Nevada, Las Vegas, NV 89102, USA; kyawmin.tun@unlv.edu; 3Department of Internal Medicine, Harbor-UCLA Medical Center, Torrance, CA 90502, USA; jhong5@dhs.lacounty.gov; 4Department of Medical Education, Kirk Kerkorian School of Medicine, University of Nevada, Las Vegas, NV 89102, USA; kavita.batra@unlv.edu; 5Office of Research, Kirk Kerkorian School of Medicine, University of Nevada, Las Vegas, NV 89102, USA

**Keywords:** fecal microbiota transplant, decompensated cirrhosis, *Clostridioides difficile* infection, hepatic encephalopathy

## Abstract

**Background and Aims:** Due to increasing knowledge of the “gut–liver axis”, there has been growing interest regarding the use of fecal microbiota transplant in the management of chronic liver disease. There are limited data available and current guidelines are mostly based on expert opinions. We aim to perform the first systematic review investigating safety and efficacy of fecal microbiota transplant particularly among high-risk decompensated cirrhosis patient populations. **Methods**: Literature search was performed using variations of the keywords “fecal microbiota transplant” and “cirrhosis” on PubMed/Medline from inception to 3 October 2021. The resulting 116 articles were independently screened by two authors. In total, 5 qualifying studies, including 2 randomized control trials and 3 retrospective case series, were found to meet established eligibility criteria and have adequate quality of evidence to be included in this review. **Results**: Of the total 58 qualifying patients, there were 2 deaths post fecal microbiota transplant, 1 of which could not rule out being related (1.7%). Among the remaining 56 participants, 8 serious adverse events were reported, of which 2 could not rule out being related (3.6%). The success rate of fecal microbiota transplantation in treating recurrent *Clostridioides difficile* infection among patients with decompensated cirrhosis was 77.8%. The success rate when used as investigational treatment for hepatic encephalopathy was 86.7%, with multiple studies reporting clinically significant improvement in encephalopathy testing scores. **Conclusions**: We found a marginally higher rate of deaths and serious adverse events from fecal microbiota transplant in our patient population compared with the average immunocompetent population, where it was previously found to have 0 deaths and SAE rate of 2.83%. The efficacy when used for recurrent *C.*
*difficile* infection was 77.8% and 87% in the decompensated cirrhotic and general populations, respectively. Studies on efficacy in novel treatment of hepatic encephalopathy have been promising. This study concludes that fecal microbiota transplant use in decompensated cirrhosis patients should be used with caution and preferably be limited to research purposes until better data are available.

## 1. Introduction

Fecal microbiota transplantation (FMT) is a novel and growing field with expanding therapeutic potential. Limitations to widespread FMT use despite promising outcomes is likely due to the lack of large, well-designed studies; one such example is the use of FMT in cirrhotic patients. In recent years, the therapeutic role of FMT in the cirrhotic population has become more prominently discussed, as new research hypothesizes that underlying intestinal dysbiosis may play a fundamental role in worsening clinical status in such patients [1,2,3,4,5]. Although the pathogenesis is still unclear, new research hypothesizes that “portal hypertension, reduced secretion of gastric acid, impaired gastrointestinal motility, and local and systemic immunological dysfunction” are the primary factors contributing to gut microbiota derangements that lead to increase in pathogenic bacteria and decrease in commensal bacteria [2]. This further causes inflammation and damage along the intestinal membrane, allowing for translocation of bacteria and bacteria-derived byproducts such as ammonia which can further trigger liver injury and systemic inflammation and can lead to decompensation in cirrhotic patients such as hepatic encephalopathy [3]. The term “gut–liver axis” represents the intricate relationship and co-dependence between the gut microbiome and the liver [1]. By altering the composition of the gut microbiota, FMT can potentially prevent, treat, and/or curtail the severity of the decompensation event. Increase in research efforts in and improved understanding of the gut–liver axis has led to renewed discussion about the potential therapeutic uses of FMT in patients with chronic liver disease.

Yet, before further large studies can be undertaken it is imperative to determine the safety of FMT in this patient population, specifically in decompensated cirrhotics who have the highest associated risk and mortality [4]. There is currently very limited published data and no systematic reviews regarding the safety of FMT in decompensated cirrhotics as these patients are often excluded from most FMT studies due to safety concerns. In the joint British Society of Gastroenterology (BSG) and Healthcare Infectious Society (HIS) guidelines—published in 2018 and considered the most comprehensive evidence-based recommendations for practicing clinicians—the authors concluded that “FMT should be offered with caution to patients with *Clostridioides difficile* infection (CDI) and decompensated chronic liver disease” with very low grade of evidence and weak strength of recommendation [6]. In other words, any estimate of effect is uncertain due to lacking not only published evidence but even consensus among multidisciplinary expert opinions [6]. Recently, our research team conducted a systematic review that found that FMT can be a safe and effective treatment of hepatic encephalopathy (HE) and CDI in cirrhotic patients [7]. However, the review did not delineate between patients with compensated and decompensated cirrhosis [7]. Hence, in this study, we aim to perform the very first systematic review regarding safety and efficacy outcomes of FMT in decompensated liver disease.

## 2. Methods

### 2.1. Search Strategy

We performed a comprehensive literature search using variations of the keywords “fecal microbiota transplant” and “cirrhosis” including MeSH terms to identify original studies published in PubMed/Medline from inception through 3 October 2021. The systematic review was performed on studies published prior to 3 October 2021. Results were limited to human studies published in English. There was a total of 116 studies for review. See Appendix A for detailed search terms.

### 2.2. Eligibility Criteria

Inclusion criteria: (1) FMT for various indications; (2) decompensated cirrhosis defined as stage 4 fibrosis with Child–Turcotte–Pugh (CTP) C, or a major complication including variceal hemorrhage, large ascites, spontaneous bacterial peritonitis (SBP), hepatic encephalopathy (HE), hepatocellular carcinoma, hepatorenal syndrome, and hepatopulmonary syndrome; (3) reporting of patient data and outcomes after first fecal infusion; (4) adult patients (≥18 years old) of any sex; (5) minimum follow-up of 1 month; and (6) at least moderate quality of evidence.

Exclusion criteria: (1) individual case reports which reflect unique cases and significant bias; (2) studies available only as abstracts; (3) studies without patient data; (4) non-English studies; (5) animal studies; and (6) patients younger than 18 years of age.

### 2.3. Study Selection and Data Extraction

An initial 116 articles were retrieved from PubMed. Two authors (A.S.H. and K.M.T.) independently reviewed these titles and abstracts, after which 13 articles were deemed relevant with patient data. Full texts were then reviewed, after which 5 remaining studies fulfilled complete eligibility criteria. In cases of disagreement, a third reviewer (J.M.H.) arbitrated the final decision for inclusion. Study selection process by Preferred Reporting Items for Systematic Reviews and Meta-Analyses (PRISMA) statement is detailed in Figure 1. A summary of included studies is shown in Table 1, while excluded articles are listed in Appendix A. IRB review was not required as all data were extracted from published literature and no patient intervention was directly performed.

### 2.4. Study Outcomes

The primary endpoint was safety of FMT in decompensated cirrhosis patients in terms of death and serious adverse events (SAE) occurring within 30 days after FMT. SAEs were defined as death or any event requiring hospitalization. Adverse events were further determined to be more likely or unlikely to be related to FMT. Thirty days was chosen as the latest time an adverse event could still be reasonably attributed to the original fecal transplantation [7].

The secondary endpoint was the efficacy of FMT. When used for CDI, FMT success was defined as the absence of symptoms (≥3 loose bowel movements in ≤24 h) with positive CDI confirmatory stool testing within 8 weeks of treatment [13]. When used for HE, FMT success was defined by no recurrence of HE at follow-up and improvement in scores on objective testing (Encephalapp Stroop test or Psychometric Hepatic Encephalopathy Score (PHES)). PHES has previously been established to be the gold standard for the diagnosis of HE [14]. It tests for skills such as attention, and psychomotor function [15]. It is composed of five component tests: number connection test-A, number connection test-B, serial dotting test, line tracing test, and digit symbol test [14]. A score lower than −4 has been found to be associated with HE and poorer cognitive performance [14]. EncephalApp Stroop test is a validated App-based version of the Stroop test [15]. Similarly to PHES, the Stroop test also evaluates cognitive flexibility and psychomotor speed [16]. It consists of two stages: the “ON” stage where the participant names the color of the pound signs, and the “OFF” stage where the participant states the color of a discordant word [16]. If the time taken for both ON and OFF stages is longer than 269.8 s, it has been determined with sufficient sensitivity and specificity that a diagnosis of HE can be made [15,16].

We then referred to and compared our outcomes with those from previously published articles that studied the efficacy and safety profiles of FMT in various populations such as general population, immunocompetent cohort, immunocompromised patients, and cirrhotic patients.

### 2.5. Quality Assessment

Revised Cochrane risk-of-bias tool (RoB 2) was used to evaluate the methodological quality of randomized controlled trials (RCT). RoB 2 is a revised version of the original Cochrane risk-of-bias tool that has been widely used in systematic reviews [17]. The tool consists of five domains: randomization process, derivations from intended interventions, missing outcome data, measurement of the outcome, and selection of the reported result. The overall risk of bias for each RCT is determined high, low, or some concern based on the individual elements in the 5 domains [17].

For case series, observational and cohort studies, appraisal of quality and risk of bias was performed by a series of quality assessment tools developed by US National Heart Lung and Blood Institute (NHLBI) of National Institutes of Health (NIH) (https://www.nhlbi.nih.gov/health-topics/study-quality-assessment-tools) (accessed on 14 March 2022). A set of question items with Yes/No answers were used, with a “Yes” counting as a score of 1 and a “No” as a score of 0. In the tool used for case series, there were a total of 9 questions. A score of 7–9 corresponds to good quality, while scores of 4–6 and 1–3 indicate moderate and poor quality, respectively [18]. On the other hand, for observational and cohort studies, there were 14 items in total [18]. However, three items were not applicable to the studies included in our systematic review. Out of the available 11 points, studies that score 7–11, 4–6, and 1–3 were graded as good, moderate, and poor quality, respectively [18]. In the final selection stage, only studies with at least moderate level of evidence were included. Quality appraisal was performed by two independent authors (K.M.T. and K.B.). If there was any disagreement, a third reviewer (A.S.H.) evaluated the article and achieved consensus through discussion. See Appendix A for quality assessment scores for each study.

## 3. Results

Table 1 presents a summary of the 5 multi-patient studies included for final analysis in this systematic review, composed of 2 randomized control trials and 3 retrospective case series with a total of 58 patients [8,9,10,11,12]. All had decompensated cirrhosis as defined above and received FMT. Etiology of cirrhosis ranged from alcohol, chronic Hepatitis C virus, metabolic associated fatty liver disease, primary sclerosing cholangitis, and others. In 28 patients, FMT was given per guideline for recurrent/severe CDI; however, only 27 were included in the review of efficacy. One patient expired seven days later from cholangitis which did not meet our inclusion criteria of evaluating the efficacy of FMT within 8 weeks. This patient, however, was still included in the evaluation of the safety profile of FMT. In 30 patients, FMT was given for refractory HE as an investigational therapy. Two studies were from the United States [8,9], one each from India [11] and Spain [12], and one was a multinational study from the United States, Canada, and Italy [10]. In the study by Cheng et al., 24 of the 63 patients were considered to have decompensated cirrhosis. Thus, the overall total number of patients in our review was 58 patients. The total number of patients included in the efficacy review was 57, while that for safety review was 58. Multiple FMT delivery methods are used, including via nasogastric tube (*n* = 1), enema (*n* = 10), colonoscopy (*n* = 13), and capsule (*n* = 10). Cheng et al. described the methods of FMT administration as capsule (*n* = 3), colonoscopy (*n* = 59), and percutaneous endoscopic gastrostomy (PEG) tube (*n* = 1) but did not differentiate between cirrhotic and decompensated cirrhotic patients.

Table 2 describes adverse outcomes. There was a total of 2 deaths post FMT, 1 of which could not exclude being related to FMT (1/58). This occurred in a patient with known choledocholithiasis and recurrent cholangitis who developed severe cholangitis 7 days post FMT. Although it is difficult to determine if the cause of death was related to FMT or the patient’s prior history, translocation of bacteria from FMT could be theoretically possible. The second death was a patient with diagnosis of alpha 1 antitrypsin deficiency who developed bronchopneumonia 2 months post FMT, which was deemed to be unrelated to FMT. Among the remaining 56 participants, 8 SAE were reported of which 2 could not rule out relation to FMT (2/56). One patient developed *Escherichia coli* bacteremia 3 days post FMT without any other documented causes, which is likely to be related to FMT. One patient developed SBP in week 4, which was considered to still be possibly related to initial FMT. The remaining reported 4 SAEs were deemed unlikely to be related to FMT, including SBP at week 8, bleeding portal hypertensive gastropathy at day 23, hospitalization for HE at day 56, acute renal failure at day 85, and chest pain at day 1115 post FMT.

Table 3 shows the secondary outcome of FMT efficacy. Of the 27 decompensated cirrhotic patients who received first time FMT for severe or recurrent CDI and were followed for at least 8 weeks, 6 patients had recurrence of CDI at 8-week follow up (success rate 21/27 = 77.8%). One patient received FMT for CDI resulting in death from cholangitis at 2 weeks, and was excluded from the efficacy review as adequate follow-up endpoint was not met. Of the 30 patients who received FMT for HE, 4 patients had recurrence of HE at time of follow-up, which did range from 20 weeks to 5 months (success rate 26/30 = 86.7%). Of note, one of these patients receiving FMT for HE also resulted in death at 2 months which was deemed adequate for efficacy review. In Bajaj 2017, FMT patients had significantly fewer HE episodes (0% vs. 50%; *p* = 0.03) and PHES improvement (−3.1 vs. 0.00; *p* = 0.01) at 5 months compared to patients receiving standard of care (SOC), with PHES score improvement [8]. In Bajaj 2019, patients who received FMT had a clinically significant improvement in Encephalapp performance compared to those receiving SOC (*p* = 0.02) [9]. In Mehta 2018, patients with HE receiving FMT had a statistically significant reduction in CTP score 9.5 (9–10.75) vs. 8 (7–8) and MELD 18 (16.25–19) vs. 15 (14–16) [11].

## 4. Discussion

In our study, we found that the overall rate of death possibly attributable to FMT was 1.72% (1/58) with the rate of other SAEs attributable to FMT as 3.57% (2/56). In contrast, the systematic review on cirrhotic patients by Tun et al. demonstrated that overall death rate was 1.57% (2/127), and the number of SAEs and AEs were 9.45% (12/127) and 19.68% (25/127), respectively [7]. The data suggests that hepatic decompensation may not be correlated with or attributable to SAEs or AEs. We compared our data with a meta-analysis conducted by Michailidis et al. that studied high quality randomized controlled trials and included all patients receiving FMT without factoring in comorbid conditions to represent the baseline population [19]. In this study, death had to be directly attributed to FMT or peri procedural [19]. Serious adverse drug experiences were defined as those that result in death, a life-threatening adverse drug experience, inpatient hospitalization or prolongation of existing hospitalization, a persistent or significant disability/incapacity, or a congenital anomaly/birth defect during the follow up 30 days after FMT [19]. This study of the general population found 0 deaths (0/388) and rate of total SAEs to be 2.83% (11/388) which is comparable in terms of safety data to our study population [19]. A separate large but single center study was conducted by Youngster et al., regarding use of FMT to treat CDI in the general population [20]. There were no SAEs; AEs were reported in 30% of patients [20]. In a meta-analysis by Shogbesan et al. of 44 non-randomized studies of FMT in an immunocompromised population (patients on immunosuppressant medications, with human immunodeficiency virus (HIV), inherited or primary immunodeficiency syndromes, cancer undergoing chemotherapy, or organ transplant, including bone marrow transplant), the overall rate of death was 0.66% (2/303) due to aspiration and pneumonia; rate of serious adverse events leading to hospitalization was 8.3% (25/303) [21]. The authors concluded that the overall rates of SAEs in the immunocompromised patients were similar to those in the immunocompetent population [21]. Overall, the rate of death in our study population was slightly higher compared to that in Shogbesan’s study population, although the rate of SAEs remained lower.

In terms of efficacy data for CDI, we found that the rate of success for first time FMT treatment in the decompensated cirrhotic patient population was 77.8% (21/27), which is lower than previously reported data such as a randomized control trial by Cammarto et al. where efficacy was 90% [22]. Of note, in Shogbesan’s study of immunocompromised patients, the efficacy was found to be 207/234 (87%), comparable to a normal immunocompetent population [21]. Therefore, the overall efficacy of first FMT in CDI in our patient population was lower than the general as well as the immunocompromised population, although it still remains high overall. In comparison, success rate for treatment of CDI among cirrhotic patients was 86% (82/95) [7]. The lower rate of cure in our study indicates that decompensation may hinder the therapeutic potency of FMT in treating CDI. Such patients may need multiple doses of FMT for resolution of CDI. The findings from our study are also consistent with the results from Shogbesan’s study; it was discovered that patients with a single immunocompromising element achieved a higher success rate of treatment than those with multiple immunodeficiencies (*p* < 0.001) [21]. It has been noted that a patient’s immune status may determine the number of doses required or the success rate for FMT [7]. Indeed, Shogbesan et al. discovered that immunocompromised patients required more than one dose of FMT to achieve cure and that the success rate improved from 88% to 93% with multiple FMTs [21]. With cirrhosis already being an immunocompromising factor, decompensation may exacerbate the deficiencies in the immune status and may lessen the efficacy of FMT. This could be a reason that the decompensated cirrhotic patient population had a lower success rate than the cirrhotic patients, other immunocompromised patients, or the general population. Hence, it is crucial to take decompensation into consideration while considering FMT for treatment of CDI and to counsel patients on potential reduced effectiveness and/or need for multiple administrations. It may be necessary to address the decompensation prior to proceeding with FMT.

FMT also appears to be effective in HE. Our review demonstrates that an improvement in cognitive tests (PHES and EncephalApp Stroop test) or reduced recurrence of HE was seen across all the studies compared to SOC. Furthermore, the effect of FMT was maintained over the course of the year. Similar findings were reported in the previous systematic review on cirrhotic patients [7]. A similar finding was corroborated by Bloom et al. Among cirrhotic patients with HE, PHES improved by 2.1 points after three doses of FMT and by 2.9 points after five doses [23]. Furthermore, overall PHES remains improved by 3.1 points 4 weeks after the fifth dose of FMT [23]. In comparison, improvement in the Stroop test was seen only with five doses of FMT [23].

## 5. Limitations

The primary weakness of this study is due to the lack of published literature with direct patient data on FMT use in cirrhosis patients. Due to the limited number as well as the fact that efficacy was often reported but was not the primary endpoint in these studies, a true meta-analysis was unable to be performed. Two of our strongest studies included were by the same research group, and although two separate selection processes of patients were used and made up most of the data for use of FMT in treating HE. Due to the nature of this systematic review, we were unable to control for confounding factors or variables both in the patient qualities and in the procedural protocols (method of FMT, type of FMT given). Additionally, AEs and SAEs were reported by the original authors as related or unrelated to FMT based on DSMB guidelines (Data and Safety Monitoring Board). Since we were not able to obtain the original data of the studies, it is possible that the number of adverse events attributable to FMT may be inaccurate. Lastly, after an initial search on multiple databases and exclusion of articles that did not meet the inclusion criteria, it was noted that the results were largely identical. Therefore, only PubMed/Medline was used to retrieve qualifying articles for further review as it is the largest and most robust database, which may not have extracted smaller studies and may have introduced publication bias.

## 6. Conclusions

Based on available literature, we found that there may be a marginally higher rate of death and SAE from FMT in decompensated cirrhosis compared with the average immunocompetent population. When used for CDI, the efficacy of one-time infusion was also found to be less effective. New data does appear to support a therapeutic advantage of FMT compared with SOC in hepatic encephalopathy. We conclude that although FMT is likely to be a valuable tool in the future management of cirrhosis patients, its use in decompensated cirrhosis patients should be used with caution and preferably limited to research purposes until better safety and efficacy data is available. Although this study was confined by paucity of published data and inability to perform a true meta-analysis, it does highlight the need for more robust randomized controlled studies regarding FMT in this specific patient population in the future.

## Figures and Tables

**Figure 1 antibiotics-11-00838-f001:**
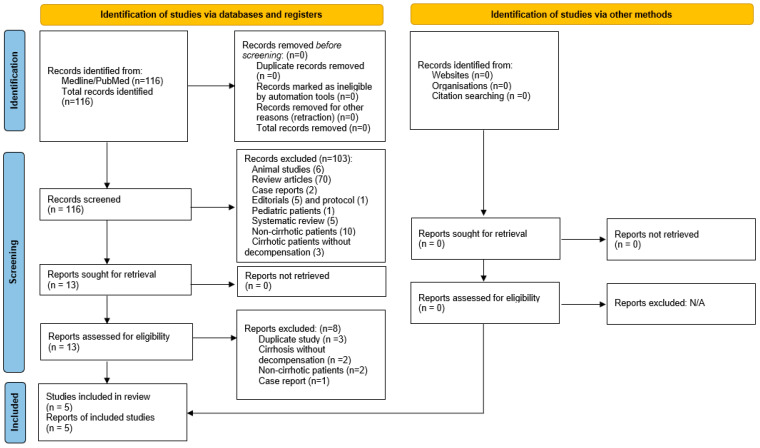
Preferred Reporting Items for Systematic Reviews and Meta-Analyses.

**Table 1 antibiotics-11-00838-t001:** Summary of included studies.

Author/ Year	Study Design	Quality Assessment	Quality Score	Dates	Sample Size	# Pts FMT	Follow-up Period	Pt qualifications	Etiology Cirrhosis	Exclusions	Indication for FMT	FMT Method
Bajaj 2017 * [8]	RCT	RoB 2	8	10/2015–7/2016	20	10	5 months	Age > 18, cirrhosis with recurrent HE at least 2 documented episodes requiring therapy	Hepatitis C virus, Non-alcoholic fatty liver disease, alcohol, others	MELD > 17, allergies to pre FMT abx, antimicrobials, immunosuppressive, CDI, pregnancy, EtOH, unable to give informed consent	HE	Enema
Bajaj 2019 * [9]	RCT	RoB 2	8	7/2017–5/2018	20	10	5 months	Cirrhosis w recurrent HE at least 2 episodes within last year on lactulose and rifaximin	Hepatitis C virus, Non-alcoholic steatohepatitis, alcohol, others	MELD > 17, unable consent, current abx, contraindication to endoscopic procedure	HE	Open-biome capsule
Cheng 2020 [10]	Retro-spective study	NIH quality assessment	9	1/2012–11/2018	63	24 ^#^	12 weeks	Decompensated cirrhosis (ascites, varices, variceal hemorrhage, HE)	Hepatitis C virus, alcohol, Non-Alcoholic SteatoHepatitis, others	Patients with <12 week follow up and history of liver transplantation	Recurrent, severe CDI	Capsule Colonoscopy PEG
Mehta 2018 [11]	Case series	NIH quality assessment	7	8/2017–10/2017	10	10	20 weeks	Hepatic encephalopathy more or equal to 2 episodes of grade 2–4 HE in last 6mo	Alcohol, Non-alcoholic steatohepatitis, Hepatitis C virus	Active EtOH, positive CDI, on immunosuppressive or antimicrobial	HE	Colonoscopy
Olmedo 2019 [12]	Case series	NIH quality assessment	8	2013–2017	4	4 **	4–11 months	Cirrhosis (CP C or esophageal varices bleeding) and CDI	Alcohol, Hepatitis C virus	None	Recurrent severe CDI	Colonoscopy or NG tube

RCT = Randomized controlled trial; RoB 2 = Revised Cochrane risk-of-bias tool; HE = Hepatic encephalopathy; MELD = Model for End-Stage Liver Disease; FMT = Fetal microbiota transplantation; CDI = *Clostridioides difficile* infection; EtOH = Ethanol; NIH = National Institutes of Health; PEG = Percutaneous endoscopic gastrostomy; NG = Nasogastric. * Bajaj 2017 and Bajaj 2019 patient data is published in multiple other follow up studies. The first published versions were chosen to be included in this study as they were adequate to fulfill this study’s eligibility requirements and provided most information regarding the study population’s cirrhosis status. # Although the total number of patients who received FMT in Cheng 2020 was 63, only 24 patients were diagnosed with decompensated cirrhosis. ** One patient expired before the efficacy profile associated with FMT could be evaluated.

**Table 2 antibiotics-11-00838-t002:** Adverse event outcomes.

Study	Death	SAE	AE	Unrelated
Bajaj 2017 [8]	None	None	Unknown	2– day 85 for AKI and day 1115 for chest pain was neg ACS
Bajaj 2019 [9]	None	None	1 UTI from *Klebsiella pneumoniae* 2 months post; 1 pneumonia and receiving alpha 1 antitrypsin infusions	1– post TIPS complication HE not related to FMT
Cheng 2020 [10]	None	None	Unknown	2– bleeding portal hypertensive gastropathy 23 days after FMT; hepatic encephalopathy 56 days after FMT
Mehta 2018 [11]	1– bronchopneumonia 2 months after FMT	1– SBP at week 4	Unknown	1– SBP at week 8
Olmedo 2019 [12]	1– death 7 days post FMT from cholangitis	1– *Escherichia. Coli* bacteremia 3 days post FMT without other cause	Unknown	None

SAE = Serious adverse events; AE = Adverse events; AKI = Acute kidney injury; ACS = Acute coronary syndrome; UTI = Urinary tract infection; TIPS = Transjugular intrahepatic portosystemic shunt; HE = Hepatic encephalopathy; FMT = Fecal microbiota transplantation; SBP = Spontaneous bacterial peritonitis.

**Table 3 antibiotics-11-00838-t003:** Efficacy outcomes.

Study	Indication for FMT	Definition of efficacy	Outcome
Bajaj 2017 [8]	HE	No recurrence of HE	significant less HE episodes at 5 months (0% vs. 50% *p* = 0.03); PHES score improvement (−3.1 vs. 0.00 *p* = 0.01); MELD score no clinically significant difference (0.78)
Bajaj 2019 [9]	HE	EncephalApp and no recurrence of HE	EncephalApp performance improved post FMT only (*p* = 0.02); 3 patients had no recurrence of HE. 1 patient had HE recurrence
Cheng 2020 [10]	Recurrent, severe, fulminant CDI	No recurrence of CDI	18 out of 24 patients with decompensated cirrhosis who received FMT had resolution of CDI. 6 patients had recurrent CDI at follow up
Mehta 2018 [11]	HE	No recurrence of HE, CTP, MELD	7 out of 10 patients had no recurrence of HE; statistically significant reduction in CTP score (9.5 9–10.75) vs. 8 (7–8) and MELD 18 (16.25–19) vs. 15 (14–16)
Olmedo 2019 [12]	Recurrent severe CDI	Not well defined	3 out of 3 patients had resolution of CDI. 1 patient expired from cholangitis within 7 days of FMT and was excluded from efficacy review

FMT = Fecal microbiota transplantation; HE = Hepatic encephalopathy; PHES = Psychometric Hepatic Encephalopathy Score; MELD = Model for End-Stage Liver Disease; CDI = *Clostridioides difficile* infection.

## Data Availability

Data supporting the statements can be found on PubMed database.

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
