# Peer review of "Fecal Microbiota Transplantation in Decompensated Cirrhosis: A Systematic Review on Safety and Efficacy"

_antibiotics, 2022, doi:10.3390/antibiotics11070838_

Round 1

Reviewer 1 Report

Thank you for the opportunity to review your systematic review.

Title: ‘Fecal Microbiota Transplantation in Decompensated Cirrhosis: A Systematic Review on Safety and Efficacy’

The review provides essential outcomes on the usage of FMT. It was a well-written paper considering PRISMA guidelines. However, some questions remain unanswered or need clarification.

Major comments:

Inclusion criteria indicate patients of any age and sex. It would be better to clarify why you excluded pediatric patient records in the PRISMA flow chart. It might impact your outcomes.

It would be better to mention the reason for search restriction (e.g., only used Pubmed).

It would be better to explain why not studying the comparator or control. Please consider the PICOT method.

Minor comments:

It would be better to add more info on “gut-liver axis’ in the introduction to understand the rationale of the systematic review.

Line 23-26: Not clear enough to the readers; please reword those sentences. E.g., The success rate among this study population when considering the non-recurrent Clostridioides difficile infection was 77.8% or better words…

Line 55: It would be better to provide references on often excluded cirrhotic patients from FMT studies.

Line 64: Please provide the full term of abbreviated HE when it appears the first time. Line 79: Please remove the full term of HE.

Line 103: It would be better to add a supportive reference.

It would be better to indicate the review ended date.

Reviewer 2 Report

In the present systematic review Hong et al summarized the main studies investigating the role of fecal microbiota transplantation (FMT) in cirrhotic patients for C. difficile infection and hepatic encephalopathy. Main comments:

1) Page 2 line 47: what do Authors mean for “gut border”?

2) The Newcastle Ottawa scale should be used for nonrandomized studies. It is incorrect in this setting.

3) Authors affirm that the total amount of enrolled patients was 58, while in the study by Cheng (ref. 16), 63 subjects were recruited. Please explain.

4) I suggest to repeat and update literature search, since some reports may not have been included >(see Sharma A et al, Hepatol Int 2022).

5) Please discuss briefly the role of gut microbiota in decompensated cirrhosis and how FMT could impact on the pathogenesis of encephalopathy and other complications.

Round 2

Reviewer 2 Report

NOS scale CANNOT be applied to randomized trials. Please use Cochrane tool.

All other answers were OK.

Author Response

Thank you for your comment. As you suggested, we switched the quality assessment tool from NOS to Cochrane tool. We used the Revised Cochrane risk-of-bias tool (RoB 2) since it is the updated version of the widely used Cochrane-risk-of bias tool. We used it for the studies Bajaj 2017 [14] and Bajaj 2019 [15]. We adjusted our Methods section to reflect these changes from lines 137 to 143. We also added the full assessment to the Supplementary material. References corresponding to NOS were removed and a new reference for RoB 2 was added at Reference 12.
